# Novel Low Pathogenic Avian Influenza H6N1 in Backyard Chicken in Easter Island (Rapa Nui), Chilean Polynesia

**DOI:** 10.3390/v14040718

**Published:** 2022-03-30

**Authors:** Francisca Di Pillo, Cecilia Baumberger, Carla Salazar, Pablo Galdames, Soledad Ruiz, Bridgett Sharp, Pamela Freiden, Shaoyuan Tan, Stacey Schultz-Cherry, Christopher Hamilton-West, Pedro Jimenez-Bluhm

**Affiliations:** 1Núcleo de Investigaciones Aplicadas en Ciencias Veterinarias y Agronómicas, Facultad de Medicina Veterinaria y Agronomía, Universidad de Las Américas, Santiago 7500978, Chile; fdipillo@udla.cl (F.D.P.); soleruizp@gmail.com (S.R.); 2Departamento de Medicina Preventiva Animal, Facultad de Ciencias Veterinarias y Pecuarias, Universidad de Chile, Santiago 8820808, Chile; cecilia.baumberger@uchile.cl (C.B.); carlasll@veterinaria.uchile.cl (C.S.); pgaldames@veterinaria.uchile.cl (P.G.); 3Department of Infectious Diseases, St. Jude Children’s Research Hospital, Memphis, TN 38105, USA; bridgett.sharp@stjude.org (B.S.); pamela.freiden@stjude.org (P.F.); shaoyuan.tan@stjude.org (S.T.); stacey.schultz-cherry@stjude.org (S.S.-C.); 4Escuela de Medicina Veterinaria, Facultad de Agronomía e Ingeniería Forestal, Facultad de Ciencias Biológicas y Facultad de Medicina, Pontificia Universidad Católica de Chile, Santiago 7820436, Chile

**Keywords:** influenza A, avian influenza, backyard chicken, Rapa Nui, Easter Island, Polynesia

## Abstract

Little is known about the prevalence of avian influenza viruses (AIV) in wildlife and domestic animals in Polynesia. Here, we present the results of active AIV surveillance performed during two sampling seasons in 2019 on Easter Island (Rapa Nui). Tracheal and cloacal swabs as well as sera samples were obtained from domestic backyard poultry, while fresh faeces were collected from wild birds. In addition to detecting antibodies against AIV in 46% of the domestic chickens in backyard production systems tested, we isolated a novel low pathogenic H6N1 virus from a chicken. Phylogenetic analysis of all genetic segments revealed that the virus was closely related to AIV’s circulating in South America. Our analysis showed different geographical origins of the genetic segments, with the PA, HA, NA, NP, and MP gene segments coming from central Chile and the PB2, PB1, and NS being closely related to viruses isolated in Argentina. While the route of introduction can only be speculated, our analysis shows the persistence and independent evolution of this strain in the island since its putative introduction between 2015 and 2016. The results of this research are the first evidence of AIV circulation in domestic birds on a Polynesian island and increase our understanding of AIV ecology in region, warranting further surveillance on Rapa Nui and beyond.

## 1. Introduction

Avian influenza viruses (AIV) remain a continuous threat to both human and animal populations, playing an important role in different outbreaks over the past century [1]. AIVs belong to the Alphainfluenzavirus (Influenza A) genus of the Orthomyxoviridae family, which can be divided into subtypes based on the haemagglutinin (HA) and neuraminidase (NA) antigens. To date, 16 HA subtypes (H1–H16) have been described in avian reservoirs, from which H5 and H7 viruses can become highly pathogenic in chicken and turkeys [2]. Particularly, Anseriformes and Charadriiformes orders act as natural reservoirs and can spread the virus between continents during migrations [3].

Despite recent efforts [4,5,6,7,8], South America remains as one of the most neglected areas in the world in terms of AIV surveillance. Studies in Argentina, Chile, and Peru have shown the circulation of a South American AIV lineage, indicating a divergent evolution of these viruses in the Southern hemisphere [4,6,7]. In contrast, AIV from wild birds in Chile have a wide variety of subtypes and reassortant viruses containing genes from both North American and South American lineages [6]. Evidence also points at a widespread circulation of AIV in backyard production systems (BPS) in Chile [9,10,11], which could be the result of spillover events from wild birds [10].

Rapa Nui or Easter Island is part of the Chilean insular territory and the Polynesian Triangle, located 3790 km west from the coast of Chile, 2075 km east from the Pitcairn islands, and 4351 km southeast from Tahiti, being one of the most isolated inhabited islands in the world. It has a subtropical weather, with year-round rainfall. According to the official results of the last census carried out in 2017 by the National Statistics Institute, Rapa Nui has 7750 inhabitants in an area of 163.6 km^2^ and is visited by over 100,000 tourists a year [12].

Domestic chickens (*Gallus gallus*) play an important role on the island [13], acting as a food source when seafood was scarce [14], and for their feathers, which are used in decorative objects such as crowns and costumes for rituals [15]. The chickens are found around the houses, mainly kept in BPS, where they can be seen roaming freely during the day and locked in chicken coops at night [13]. According to data published in Marin and Caceres [15], there were many introductions of chickens from the mainland, particularly by missionary priests between 1940 and 1950. However, resolution 2446 (Resolution 2446/1993. Department of agriculture. Agricultural and Livestock Service. Available online: http://www.sag.cl/sites/default/files/resol.2446_de_1993-isla_de_pascua.pdf, accessed on 15 November 2021) of the Ministry of Agriculture issued in 1993 prohibits the entry of domestic animals to Rapa Nui. Hence, the domestic bird populations of the island should be mostly naïve to pathogens circulating worldwide after 1993.

While one of the main characteristics of Rapa Nui is its extreme geographical isolation, which would suggest a physical barrier to the transmission of infectious agents, there is a large influx of tourists from all over the world, in addition to trade by ship and air as well as to migratory birds, that could contribute to the introduction of AIV to the island. Interestingly, Rapa Nui does not fall directly under any of the closest migratory flyways that connect the northern and southern hemispheres, i.e., the West-Pacific and Pacific-American Flyways; hence, many of the birds that visit the island are tropical pelagic seabirds (terns, frigatebirds, noddies, boobies, and petrels) [16]. Furthermore, excluding Australia and New Zealand, there is no scientific evidence describing the presence of AIV on islands belonging to the Polynesian triangle [17,18,19]. Therefore, this study’s objective was to identify the presence of influenza A in backyard poultry and wild birds in Rapa Nui, as well as to characterize their serologic and molecular attributes.

## 2. Materials and Methods

### 2.1. Ethics Statement

All sampling activities were approved by the Institutional Animal Care and Use Committee (CICUA) of the University of Chile.

### 2.2. Sampling and Study Area

All poultry samples were taken at BPS located in Rapa Nui, in and around the town of Hanga Roa (27.14 latitude South and 109.42 longitude West). Sampling was carried out in March 2019 and December 2019. Due to the absence of any BPS registries, a convenience sampling strategy was implemented that included all poultry species; all animals on the BPS were sampled. Samples consisted of tracheal and cloacal swabs, as well as sera samples. Cloacal and tracheal flocked swabs (Copan group, Brescia, BS, Italy) were stored in 1 mL Universal Transport Media (Copan group, Brescia, BS, Italy) and stored at 4 °C for 5 days until reaching the Faculty of Veterinary Medicine of the University of Chile, where they were stored at −80 °C until analysis. Blood samples were collected in 5 mL serum separator tubes, centrifuged 4000 g for 15 min, and stored at 4 °C for 5 days until reaching the laboratory and then stored at −30 °C until analysis. A maximum of 3 mL blood was drawn from the ulnar or jugular veins from adult birds.

All wild bird samples were taken at Moto Nui, a 3.9-hectare uninhabited islet 1500 m (~4900 feet) west of Rapa Nui. It is one of the few accessible nesting sites of pelagic bird on Rapa Nui and is frequently visited by the Red-tailed tropicbird (*Phaethon rubricauda*) and the Pacific masked booby (*Sula dactylatra*). Eighty-four fresh wild bird feaces samples were only taken in March 2019 due to the inability to disembark in December 2019. Fresh faeces were collected with a sterile flocked swab (Copan group, Brescia, BS, Italy) and stored at 4 °C in a 1 mL Universal Transport Media (Copan group, Brescia, BS, Italy) for 5 days until reaching the laboratory, where samples were stored at −80 °C until analysis.

### 2.3. Molecular Analysis and Virus Isolation

Moreover, 50 uL of tracheal and rectal swab samples were RNA extracted using the MagMax-96 AI/ND viral RNA extraction kit (Life Technologies Corporation, Carlsbad, CA, USA) following manufacturer’s instructions. The influenza A Matrix gene was detected using the 4× Fast Virus Master Mix (Thermofisher Scientific, Waltham, MA, USA) on a Stratagene m3000p (Agilent technologies, Santa Clara, CA, USA) Real-Time PCR with primers and probes as described elsewhere [20] and a cycle cutoff value of 38 [21].

RT-qPCR positive samples were blind passaged in 10-day old embryonated chicken eggs as described [22]. Viral titres were determined by endpoint dilutions based on the Reed and Munch method [23], using 50% infectious tissue dose (TCID_50_) in Mardin-Darby canine kidney cells (MDCK).

### 2.4. Sequencing and Phylogenetic Analysis

Samples where sequenced at the St. Jude Children’s Hospital Hartwell Center on the Illumina platform on a MiSeq personal genome sequencer (Illumina, San Diego, CA, USA) using the Nextera XT DNA-Seq library preparation kit, following established laboratory procedures [24]. De novo sequence assembly was performed using the SPADes package [25] and the computational capabilities of the St. Jude High-Performance Linux Computer cluster.

Each segment was analysed separately by phylogenetic analysis. Reference sequences for phylogenetic analysis of each but the NS segment of A/Chicken/Rapa Nui/CT2170/2019 were obtained from the 100 closest sequences as determined by BLAST [26]. Reference sequences for the NS as well as additional representative sequences for other segments were obtained from the Influenza virus resource of NCBI [27]. Sequences were aligned using MUSCLE (v.3.8.31) [28] and visually inspected and trimmed in BioEdit (v.7.2.5) [29]; repeated and incomplete sequences were removed. To evaluate time-signal and to eliminate any outliers for each alignment, a heuristic regression analysis was performed on TempEst (v.1.5.1) [29]. Maximum likelihood trees for the aforementioned analysis where build with RaxML (v.8.1.2) [30] using the GTR + γ with 500 bootstrap replicates to confer statistical robustness to the tree. To evaluate the time to the most recent common ancestor (tMRCA), a time-stamped Bayesian inference was carried out on BEAST (v.1.10.4) [31]. Coalescent constant size population with a lognormal-relaxed evolutionary priors were used. Additionally, a HKY + I + γ substitution model was chosen for each of the segments [32,33,34]. At least 50 million Markov Chain Monte Carlo (MCMC) iterations were run by segment to obtain a >200 effective sample size (ESS) per parameter as observed on Tracer (v.1.7.1). For each chain, the first 10% burn-in was removed and at least three independent runs were later combined using LogCombiner (v.1.10.4). The time-stamped maximum clade credibility (MCC) trees were summarized and annotated in TreeAnnotator (v.1.10.4) and visualized in FigTree (v.1.4.3).

### 2.5. Serological Analysis

Sera were tested by NP-ELISA (Virusys Corporation, Randallstown, MD USA) and the assay was carried out following the manufacturer’s instructions. Optical density values were read on a Tecan Sunrise (Tecan Group Ltd., Männedorf, Switzerland) spectrophotometer at 450 nm. Hemagglutination inhibition assay (HAI) against A/mallard/Chile/C9763/2016 (H6N8), one of the earliest H6 isolates obtained from the central region of mainland Chile, was performed on NP-ELISA positive sera. Briefly, NP-ELISA positive sera were treated with RDE (receptor destroying enzyme) (Denka Seiken Co., Tokyo, Japan), and then 2-fold serially diluted in duplicate in 25 uL PBS on a 96 well V-bottom plate. Next, 25 uL of virus at 4 hemagglutination units were added and the plate was incubated for 30 min at room temperature. Readout was performed with 50 uL chicken red blood cells diluted in PBS at 0.5%.

## 3. Results

### 3.1. Sampling

A total of seven BPSs were visited in March 2019 and six during December 2019. The collected samples included 130 chickens, 4 domestic ducks, and 1 peacock. Overall, 135 tracheal swabs (66 in March 2019 and 69 during December 2019), 135 cloacal swabs (66 in March 2019 and 69 in December 2019), as well as 135 sera samples (66 in March 2019 and 69 in December 2019) were collected for analysis. Additionally, 84 fresh faecal samples from wild birds were collected at Moto Nui for analysis.

### 3.2. Serology

Of the 135 collected sera, 60 were NP-ELISA positive, indicating an influenza A seropositivity in poultry of 44.4% (95% CI 36.3–52.8). While not all the same, BPS were visited in both seasons, and seropositivity remained similar throughout the sampling period with 42.4% (95% CI 31.2–54.4) (*n* = 28 positives out of 66) and 46.4% (95% CI 35.1–58) (*n* = 32 positives out of 69) seropositivity, respectively. NP-ELISA-positive sera were further characterized by HAI assay. None of the NP-ELISA-positive sera reacted against A/mallard/Chile/C9763/2016 (H6N8).

### 3.3. Molecular and Phylogenetic Analysis

While all the wild bird samples and cloacal swabs from domestic chickens were negative by RT-qPCR, we did have one positive tracheal swab indicating a positivity of 0.74% (95% CI 0.13–4.0). Despite the high cycle threshold (Ct) value of the samples (36.23), A/Chicken/Rapa Nui/CT2170/2019 virus was successfully isolated after blind passaging in embryonated hen eggs. Sequences were deposited in GenBank under accession numbers MZ707537 and MZ707484 to MZ707490.

Phylogenetic analysis of the HA segment revealed it to be related to sequences obtained from wild birds in wetlands and coastal areas from the central zone of Chile between 2016 and 2017, with a tMCRA in 2013 (95% HPD 2011.9–2015.1). Pairwise sequence analysis showed a 96.1% nucleotide and 98.5% amino acid identity to the closest continental strain A/Franklins gull/Chile/C10242/2016 (H6N2) (Table 1). The long branch of the HA compared to other sequences in this clade indicates a lack of active AIV surveillance on the island and hints at an independent evolution of the pathogen (Figure 1A). Overall, the Chilean H6 viruses clade together with other H6 isolates obtained in Argentina between 2007 and 2008, with a tMRCA estimated in 2001 (95% HPD 1997.4–2004.3). The entire South American H6 clade diverged from the North American H6 clade approximately in 1982 (95% HPD 1969.8–1993.6).

Similar to the HA, the NA coding segment was related to sequences obtained from wild birds in Chile between 2015 and 2016 (Figure 1B). The tMRCA of the NA was 2014 (95% HPD 2014.3–2015.6). This subtype is closely related to other NA segments obtained in the central region of Chile, with the closest sequences isolated from dabbling ducks in 2015 [(A/yellow-billed teal/Chile/C4126/2015 (H1N1)], sharing a 98.2% nucleotide and 98.5% amino acid identity (Table 1). No NA stalk deletion was observed during sequence inspection.

The phylogenetic analysis of the internal segments indicates that A/Chicken/Rapa Nui/CT2170/2019 is related to genetic segments obtained from wild birds in Chile and Argentina, except for the PA gene, that closely resembles an environmental sample (Figure 2). Nucleotide identity of the internal genes to the closest continental strains ranged from 88.9% to 99.7% and amino acid identity from 96.8% to 100% (Table 1). Interestingly, the PB2, PB1, and NS gene segments are closely related to viruses isolated form dabbling ducks in Argentina in 2016, with the tMRCA to these sequences in 2016 (HPD 2015.8–2016.4), 2015 (95% HPD 2015.6–2016.4), and 2016 (HDP 2015.6–2016.5), respectively. Both the PB2 and the NS gene are associated with the same strain, A/yellow-billed pintail/Argentina/CIP112-1174A/2016. All internal segments of A/Chicken/Rapa Nui/CT2170/2019 had tMRCA’s to the closest continental sequences ranging between late 2015 and early 2016, except for the MP gen that showed a tMRCA closer to the end of 2016 but with a lower bound 95% HDP interval overlapping with the other segments (Table 1).

## 4. Discussion

A recent study aimed at identifying possible threats of infectious diseases to native avifauna in French Polynesia, has indicated that there is no evidence of avian influenza virus in the area [35]. The Pacific Island countries and territories (PICTs) are also reported to be free of Highly Pathogenic Avian influenza, among other serious infectious livestock diseases that are commonly prevalent in other parts of the world. However, a systematic review by Brioudes, Warner [36] has indicated that there is a lack of scientific evidence to confirm this statement. The authors showed that between 1992 and 2012, AIV had not been detected nor present in domestic poultry in Oceania. When searching for information regarding wild birds, four isolations of subtype H4N6 and two of H5N2 were made in 1997 from wild ducks during routine surveillance activities in New Zealand [18,19]. These are the only published data regarding the situation of AIV in Oceania, highlighting the gaps of knowledge and the need to enhance the surveillance of animal diseases in this region [36].

The majority of the data in Oceania come from Australia, where one study showed that AIV subtypes H6 and H9 were commonly present in the studied population [17]. However, Australia does not belong to the Polynesian triangle nor to the PICTs. Therefore, information regarding the presence of influenza viruses in Polynesia is still scarce, particularly in populations of domestic birds.

Backyard production systems are widely distributed throughout the world, with Rapa Nui being no exception. Due to the ease and low economic cost for their maintenance, domestic chickens are the most abundant animal species in these productive systems. The purpose of raising birds in these systems is not necessarily oriented towards economic sustainability, nor do they comply with strict biosecurity standards; hence, these production systems are characterized by raising multiple animals of multiple species and ages in close contact with people and with low biosecurity measures, which is why BPS have been implicated in the reception and spread of various animal diseases [37,38].

The results of this study show that almost half of the backyards visited were seropositive for influenza virus, which means that the domestic bird population in Rapa Nui has been exposed. This is corroborated with the isolation of the H6N1 virus. Phylogenetic results revealed that the isolated virus is related to sequences obtained from wild birds from the central zone of mainland Chile and Argentina, with its most probable introduction to the island being sometime between 2015 and 2016. However, being the tMRCAs of the HA and NA closer to 2013–2014 and internal genes tMRCAs 2015–2016, two separated introductions of AIV into the island, followed by genetic rearrangement, cannot be discarded nor inferred from the genomic data. Furthermore, none of the genetic segments has been previously associated to domestic poultry, hence hinting at a wild bird-domestic poultry spillover event. A similar event was reported in mainland Chile in 2013, with the introduction of an H12 wild bird origin AIV into a BPS [10].

Indeed, 53 species of birds have been documented on the Island. At present, 27 species of marine and coastal birds can be observed, from which at least 12 nests are on the Island [15]. On the island, there are also eight species of land birds, of which seven are introduced species and one, *Bubulcus ibis*, is migratory. The latter is native to Africa and today is widely distributed throughout the world [15,39]. Some of the birds that visit Rapa Nui are boreal, i.e., they reproduce in the Northern Hemisphere and travel in wintertime in search of better feeding places. Some of them can be observed every year and others sporadically, as is the case of *Pluvialis fulva*, which flies between Alaska, Asia, Australia, and Rapa Nui [39]. While none of the screened wild bird samples collected in Moto Nui were positive to Influenza A by RT-qPCR, introduction of the pathogen by migratory birds cannot be discarded.

## 5. Conclusions

In conclusion, our results indicate that AIV is actively circulating in backyard poultry in Rapa Nui and that the virus is genetically similar to contemporary South American clade wild bird origin AIV found in Chile and Argentina. Further studies are needed to understand AIV dynamics and evolution in both wild and domestic animals in Rapa Nui.

## Figures and Tables

**Figure 1 viruses-14-00718-f001:**
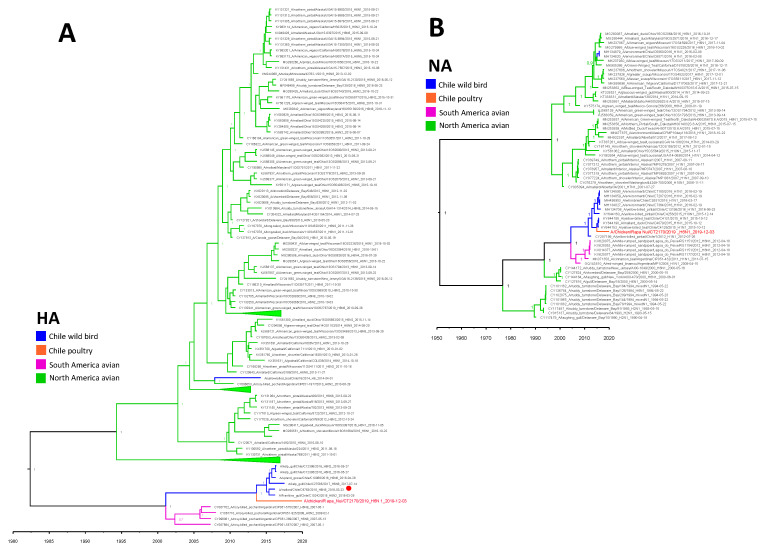
Bayesian MMC trees of the H6 (**A**) and N1 (**B**) genes. Time stamped trees that show temporal and phylogenetic relationships between the virus obtained in Rapa Nui (tip text in red) to Chilean, South American, and North American AIV’s. Node labels indicate the posterior probability of the node. Red circle in panel (**A**) indicates strain used for hemagglutination inhibition test. Collapsed nodes indicated with triangles.

**Figure 2 viruses-14-00718-f002:**
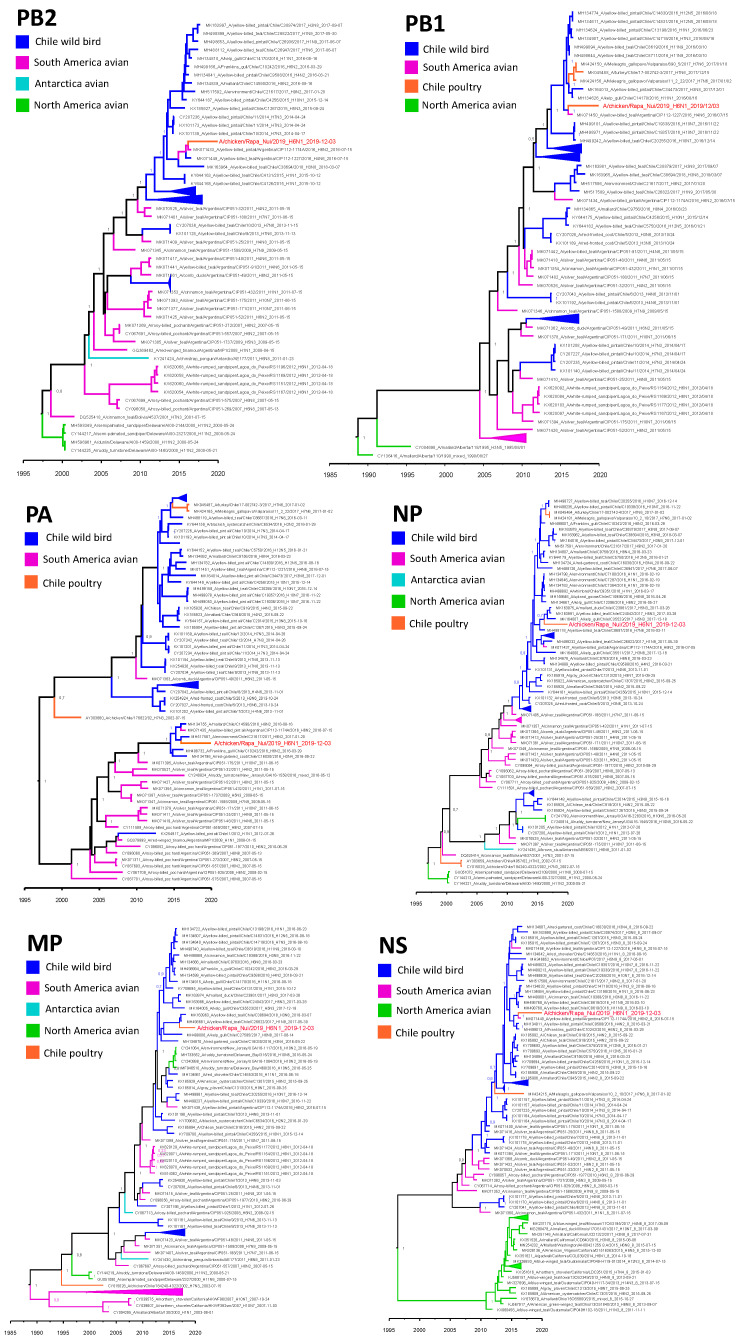
Bayesian MMC trees for each of the internal genes of A/Chicken/Rapa Nui/CT2170/2019. Time stamped trees that show temporal and phylogenetic relationships between the virus obtained in Rapa Nui (tip text in red) to Chilean, South American, Antarctician, and North American AIV’s. Node labels indicate the posterior probability of the node. Collapsed nodes indicated with triangles.

**Table 1 viruses-14-00718-t001:** Molecular characterization of the novel H6N1 virus. Closest reference sequence, time of the most recent common ancestor (tMRCA), country of isolation, nucleotide, and amino acid homology to the novel H6N1 virus shown.

Protein Coding Segment	Closest Reference Sequence	tMRCA of A/Chicken/Rapa Nui/CT2170/2019 to Closest Reference Sequence (95% HDP)	Country of Isolation	Nucleotide and Amino Acid Identity (%)	
PB2	MK071433-A/yellow-billed pintail/Argentina/CIP112-1174A/2016 (H6N2)	2016.1474 (2015.7938, 2016.49)	Argentina	98.5	100
PB1	MK071450-A/yellow-billed teal/Argentina/CIP112-1227/2016 (H4N6)	2015.9905 (2015.5587, 2016.3852)	Argentina	97.7	99.4
PA	MH517587-A/environment/Chile/C21617/2017 (H9N2)	2015.9333 (2015.2742, 2016.5603)	Chile	88.9	96.8
HA	MH499098-A/Franklins gull/Chile/C10242/2016 (H6N2)	2013.6342 (2011.8845, 2015.0703)	Chile	96.1	98.5
NP	MK163991-A/yellow-billed teal/Chile/C24042/2017 (H5N3)	2016.1022 (2015.5233, 2016.6413)	Chile	99.7	99.5
NA	KY644153-A/yellow-billed teal/Chile/C4126/2015 (H1N1)	2014.972 (2014.318, 2015.6045)	Chile	98.2	98.5
MP	MH498681-A/yellow-billed teal/Chile/C26822/2017 (H1N9)	2016.9625 (2016.3621, 2017.4081)	Chile	99.4	100
NS	MK071440-A/yellow-billed pintail/Argentina/CIP112 CIP112-1174A/2016 (H6N2)	2016.0634 (2015.5987, 2016.4783)	Argentina	98.8	99.2

## Data Availability

The data presented in this study are available on request from the corresponding author.

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
