# Peer review of "Novel Low Pathogenic Avian Influenza H6N1 in Backyard Chicken in Easter Island (Rapa Nui), Chilean Polynesia"

_viruses, 2022, doi:10.3390/v14040718_

Round 1

Reviewer 1 Report

The manuscript provides novel information about the presence of AIV in domestic chickens in Easter Island and warrants further surveillance in this and other Polynesian regions. The isolated virus appears to be related phylogenetically to sequences from wild birds in Chile and Argentina. 

It is not necessary to mention this in the manuscript but in future studies, it might be interesting to determine what the chickens eat  (see Densmore et al., 2019, Microorganisms, Sep 7(9) 334).

Suggested minor edits:

Line 14: "Influenza" - lowercase

Line 52: change "tourist" to "tourists" 

Line 81 "BPS" is spelled out previously in Line 45, does it need to be spelled out again here?

Line 116: change "Sand" to "San"

Line 130: should "where build wi" be "were built with" ?

Line 173: should "Ct" be spelled out ?

Line 259: change "form" to "from"

Line 287: insert "." at end of sentence

Line 288: change "samples" to "sample"

References: Need consistency in journal titles - some are abbreviated yet others are not, below are missing abbreviations. Similarly, need consistency in whether "periods" are used in abbreviations or not.

Line 296: Vector-Borne & Zoonotic Dis

Line 305: Transbound Emerg Dis

Lines 309 & 321: Prev Vet Med

Line 325: Arch Vir

Line 332: Am J Epidemiol

Line 336: Curr Protoc Bioinformatics

Line 337: J Mol Biol

Line 338: J Virol

Line 340:  Nucleic Acids Res

Line 346:  Bmc Evol Biol

Line 352-353: Pac Conserv Biol

Line 355: Ann N Y Acad Sci

Line 357: J Mol Genet Med

Author Response

Response Letter: Novel Low Pathogenic Avian Influenza H6N1 in backyard

chicken in Easter Island (Rapa Nui), Chilean Polynesia (viruses-1644732)

Thank you very much for your kind review of our manuscript. Below are our answers to your remarks and concerns. 

Reviewer 1

Line 14: "Influenza" – lowercase

Changed, line 14

Line 52: change "tourist" to "tourists" 

Changed, lines 52-53

Line 81 "BPS" is spelled out previously in Line 45, does it need to be spelled out again here? Deleted, thank you, line 81

Line 116: change "Sand" to "San"

Changed line 116

Line 130: should "where build wi" be "were built with"?

Changed, line 130

Line 173: should "Ct" be spelled out?

Spelled out, thank you. Line 173

Line 259: change "form" to "from"

Changed, line 260

Line 287: insert "." at end of sentence

Done, line 288

Line 288: change "samples" to "sample"

Thank you, corrected. Line 289

References: Need consistency in journal titles - some are abbreviated yet others are not, below are missing abbreviations. Similarly, need consistency in whether "periods" are used in abbreviations or not.

Line 296: Vector-Borne & Zoonotic Dis

Done, line 298

Line 305: Transbound Emerg Dis

Done, line 307

Lines 309 & 321: Prev Vet Med

Done, lines 311 and 322

Line 325: Arch Vir

Done, line 327

Line 332: Am J Epidemiol

Done, line 334

Line 336: Curr Protoc Bioinformatics

Done, line 338

Line 337: J Mol Biol

Done, line 340

Line 338: J Virol

Done, line 341

Line 340:  Nucleic Acids Res

Done, line 343

Line 346:  Bmc Evol Biol

Done, line 349

Line 352-353: Pac Conserv Biol

Done, lines 355-366

Line 355: Ann N Y Acad Sci

Done, line 358

Line 357: J Mol Genet Med

Done, line 360

Reviewer 2 Report

In this manuscript, Di Pillo and co-authors present the results of serosurveillance of avian influenza viruses in Easter Island (Rapa Nui), Chilean Polynesia. This region has not been routinely screened for the presence of avian influenza and therefore this study is of high importance and priority. The researchers collected tracheal and cloacal samples, as well as sera from domestic poultry and tested the specimens by qRT-PCR for viral RNA and NP-ELISA for anti-influenza A antibody. In addition, fecal specimens from wild birds were tested for viral RNA. There was the evidence of high seropositivity of the domestic birds, but only one live virus was isolated from a chicken. This H6N1 virus was fully sequenced and analyzed for the relatedness of each gene to other influenza viruses by a number of bioinformatics tools. In general, the manuscript is very well written and the authors provided a nice overview of the problem of avian influenza surveillance in the region, however there are some issues that need clarification prior to publication.

Major point:

  1. It is not clear why HAI was performed with only one H6N8 strain, but not with the isolated H6N1 virus and other serotypes of AIVs, such as H5 and H7.

Minor comments:

  1. Lanes 143-144: please re-phrase the sentence for better reading
  2. Lane 146: hemagglutination inhibition, not hemagglutinin inhibition
  3. Lanes 158-159. There is no need to repeat three times “(66 in March 2019 and 69 in 159 December 2019)”.
  4. Lane 180: please replace “…96.1% nucleotide 98.5% amino acid…” with “…96.1% nucleotide and 98.5% amino acid…”
  5. Lane 208: there is no need to separate table and figures with special subheading
  6. Please avoid terms “internal genes” and “internal segments”. All RNAs are internal. The meaning refers to internal proteins, not internal genes.
  7. Table 1. The title should resemble the main message of the table, not just listing the titles of all columns. For example, the title could be “Molecular characterization of the novel H6N1 virus” or something similar.

Author Response

Response Letter: Novel Low Pathogenic Avian Influenza H6N1 in backyard

chicken in Easter Island (Rapa Nui), Chilean Polynesia (viruses-1644732)

Thank you very much for your kind review of our manuscript. Below are our answers to your remarks and concerns. 

Reviewer 2

Major point:

  1. It is not clear why HAI was performed with only one H6N8 strain, but not with the isolated H6N1 virus and other serotypes of AIVs, such as H5 and H7.

This is an interesting point, however we decided not to, because of several reasons. The intention of this experiment was to infer if the virus would still be antigenically similar to the ones circulating in mainland Chile. Since we only found one subtype (H6), we decided to run the sera against the closest available isolate from mainland Chile (A/mallard/Chile/C9763/2016 (H6N8). According to our results, the circulating strain in the island (A/Chicken/Rapa Nui/CT2170/2019 H6N1) has already accumulated enough antigenic differences to render the HAI results negative. Furthermore, if our intentions were to serologically characterize the immune response of the chickens, we would have to use a potentially very large set of viruses form different geographical origins (a panel of contemporary or older H5/H6/H7/H9 viruses form Asia, North America o South America, etc.). Even then, a negative result would not mean that any of those subtypes are absent in the island, rather it could be that we did not strain-match the assay since in order to be able to interpret any negative results of the HAI test, we would need to know beforehand what the circulating strains of influenza are in the island, or at least have a good idea, which we don´t.

Minor comments:

  1. Lanes 143-144: please re-phrase the sentence for better reading

Included word “assay” for clarification of what was done following the manufacturer’s instructions, line 143.

  1. Lane 146: hemagglutination inhibition, not hemagglutinin inhibition

Thank you, changed. Line 146

  1. Lanes 158-159. There is no need to repeat three times “(66 in March 2019 and 69 in 159 December 2019)”.

Thank you for your comment, however we would like to keep it as it is, since we would need to use wording like “respectively” on each one of the samples taken to refer to the sampling date, and that can get confusing.

  1. Lane 180: please replace “…96.1% nucleotide 98.5% amino acid…” with “…96.1% nucleotide and 98.5% amino acid…”

Added “and”, line 180.

  1. Lane 208: there is no need to separate table and figures with special subheading

Deleted, line 208

  1. Please avoid terms “internal genes” and “internal segments”. All RNAs are internal. The meaning refers to internal proteins, not internal genes.

Agree, will consider for subsequent manuscripts. However, it is commonplace throughout the literature to name them this way.

  1. Table 1. The title should resemble the main message of the table, not just listing the titles of all columns. For example, the title could be “Molecular characterization of the novel H6N1 virus” or something similar.

Included your suggestion for the title of the table, thank you. Lines 209 and 211.
